# The *Digital Divide* in the Era of COVID-19: An Investigation into an Important Obstacle to the Access to the *mHealth* by the Citizen

**DOI:** 10.3390/healthcare9040371

**Published:** 2021-03-26

**Authors:** Daniele Giansanti, Giulia Veltro

**Affiliations:** 1Centre Tisp, Istituto Superiore di Sanità, 00161 Rome, Italy; 2Engineering Faculty, Tor Vergata University, 00133 Rome, Italy; giuliaveltro.94@gmail.com

**Keywords:** COVID-19, medical devices, *mHealth*, electronic surveys, digital health, digital divide

## Abstract

In general, during the COVID-19 pandemic there has been a growth in the use of digital technological solutions in many sectors, from that of consumption, to *Digital Health* and in particular to *mobile health* (*mHealth*) where an important role has been played by mobile technology (*mTech*). However, this has not always happened in a uniform way. In fact, in many cases, citizens found themselves unable to take advantage of these opportunities due to the phenomenon of the *Digital Divide* (DD). It depends on multifaceted aspects ranging from the *lack of access to instrumental and network resources*, *to cultural and social barriers* and also *to possible forms of communication disability*. In the study we set ourselves the articulated goal of developing a probing methodology that addresses the problems connected to DD in a broad sense, capable of minimizing the bias of a purely electronic submission and evaluating its effectiveness and outcome. At the moment, we have submitted the survey both electronically (with an embedded solution to spread it inside the families/acquaintances) and using the wire phone. The results highlighted three polarities (a) the coherence of the two methods; (b) the outcome of the entire submission in relation to key issues (e.g., familiarity on contact tracing Apps, medical Apps, social Apps, messaging Apps, Digital-health, non-medical Apps); (c) a *Digital Divide* strongly dependent on age and in particular for the elderly is mainly evident in the use of *mTech* in general and in particular in *mHealth* applications. Future developments of the study foresee, after adequate data-mining, an in-depth study of all the aspects proposed in the survey, from those relating to access to resources, training, disability and other cultural factors.

## 1. Introduction

The COVID-19 pandemic was characterized by unprecedented development and use of digital technologies. These in many cases have proved to be an important resource for accessing services while maintaining social distancing. In general, there has been a growth in the use of digital technological solutions in many sectors, from that of consumption, where the use of e-banking, digital transactions and online orders has become increasingly widespread, to *Digital Health* (DH) [1,2,3] and in particular to mobile health (*mHealth*) where an important role has been played by *mobile technology* (*mTech*) [4]. Citizens, in particular, found themselves receiving various offers of technological resources based on *mTech*, which in addition to the world closely linked to consumption, were concentrated in three sectors:

### 1.1. Work, School, Social Communication 

Here, *mTech* has been useful and is currently useful to support teaching, work and relational activities in an exceptional way, allowing social distancing between subjects, such as through messaging and/or video conferencing and/or social network tools [5,6].

### 1.2. Connection to Health Services 

Here, *mTech* has carried out and is carrying out the traditional role of *mHealth* in the field of *digital health* [1,2,3,4] by connecting citizens to the health system and providing them with highly innovative technological solutions.

### 1.3. New Services for Epidemiological Monitoring 

Here, *mTech* has carried out and is carrying out a specific role in this pandemic and consists of providing *mHealth* solutions for controlling and monitoring the spread of the pandemic, such as through App-based solutions for the digital contact tracing [7,8]. 

In many cases, the simple *mTech* itself has represented a real *lifebuoy* [5,6] both for the continuation of normal activities (working and teaching) [4] and for providing a safety net.

However, this has not always happened in a uniform way. In fact, in many cases, citizens found themselves unable to take advantage of these opportunities due to the phenomenon of the *Digital Divide* (DD) [9]. It depends on multifaceted aspects ranging from the lack of access to instrumental and network resources, to cultural and social barriers [10,11] and also to possible forms of communication disability. In details the DD is mainly caused by the following problems [4].

### 1.4. Access to Resources

Access to the data network limited or by the availability of resources in the region or in some cases by political reasons, such as for example due to tensions between ethnic groups and/or groups belonging to different government positions within the same state.

### 1.5. Social Factors

Due, for example, to access difficulties in disadvantaged social categories who, even for economic reasons, cannot access these technologies.

### 1.6. Cultural Factors

Even within regions with full access to technologies, uneven access to technologies was found due to cultural and training barriers. Certainly, the mobile-born, for example, have experienced a better ability to adapt than even elderly teachers and elderly doctors.

### 1.7. Disabilities

Disabilities, such as communication disabilities, which generally represent an obstacle in a non-pandemic period to access to technologies, continued to represent an obstacle even during the COVID-19 pandemic.

With effective vaccines now available, it is appropriate at this time to have some reflections about COVID-19 in relation to the *quality of life* issues [11]. DD was included [11], along with 11 others (health inequality, gender inequality, economic disadvantage, family well-being, impact on holistic well-being, economic development versus saving lives, consumption versus environmental protection, individual rights versus collective rights, international collaboration versus conflict, prevention of negative well-being, and promotion of positive well-being), among the factors affecting the *quality of life* in the COVID-19 era. While the COVID-19′s *digital health* expansion could improve the *quality of life*, the DD could exacerbate disparities [12].

## 2. Purpose of the Study

At the date of writing this article, a search on Pubmed with the two keys “COVID-19” and “*Digital Divide*” has returned 47 works, ranging on the problems listed above and well identified (this is the self-upgradable link: https://pubmed.ncbi.nlm.nih.gov/?term=digital+divide+COVID-19&sort=date).

From a look at the contributions, it also emerged that the survey tool is important and useful.

At the time of writing this study, it emerged from a research on pubmed *((COVID-19) AND (Digital Divide) AND (Survey))* that the survey tool proved to be useful for investigating:(1)The impact of bandwidth limitations [13](2)The attitude, knowledge and practice towards COVID-19 [14].(3)Learning methods [15].(4)The racial and ethnic differences in the comparisons of posts shared on COVID-19 [16].(5)The racial and ethnic differences in the areas of remote assistance during the COVID-19 pandemic [17].

From the analysis of the previous works as a whole, it is evident that DD depends in an articulated way on various factors. 

Each of these works: focused on a specific aspect (bandwidth problems, training, remote assistance, information sharing on COVID-19);does not seem to have concretely addressed the limits of administration bias through multimedia technologies which hinder the type of population affected by DD.

In light of these considerations, we have set ourselves the following objectives:(1)Propose a survey tool that addresses in an articulated way the problems that seem to be at the basis of the digital divide.(2)Propose a tool that minimizes the bias problems that may arise with purely electronic administrations (also used for social distancing)(3)Test this instrument referred to in the previous points (a, b) on a sample of subjects and also evaluate its robustness.(4)Analyze the overall results with particular reference to familiarity in the use of tools/Apps that are key elements of *mHealth* in the pandemic period.(5)Highlight the ability to categorize this familiarity into two important sub-samples represented by young and elderly people.

## 3. Materials and Methods

At the time of the pandemic, two polling methods proved appropriate for maintaining social distance.

A first method is based on the administration of electronic surveys using websites, social networks and other multimedia tools such as electronic messaging tools with *peer to peer* dissemination techniques well-established during the pandemic.

A second method is based on the administration of surveys by telephone.

In consideration of our topic, the *digital divide*, that is the difficulty of accessing digital resources, we prepared two administration anonymous solutions:The first solution is based on an electronic survey (eS) highlighting in the introduction to the receivers to spread it *peer to peer* within the domain of their acquaintances (family and friends) and supporting them in case of difficulties with the interaction with digital technology.The second solution is based on a telephone administration using the fixed network, obtaining fixed numbers from public registers randomly and requesting the same content.

This approach was aimed at minimizing bias.

We decided to use *Microsoft Forms (Microsof corporationt, Redmond, Washington, USA)* as an eS, as we had used it with success in many other applications [2]. 

Figure 1A shows the Quick Response code related to the eS with the following link. Very importantly, in the introduction (Figure 1B) of the questionnaire there is also a request to help others who are not confident in the technology to fill in the questionnaire. https://forms.office.com/Pages/ResponsePage.aspx?id=_ccwzxZmYkutg7V0sn1ZEvPNtNci4kVMpoVUounzQ3tUN0lXRkExQTVVUTdUOVdETURCNU9UN0czUy4u. 

### 3.1. Electronic Submission (ES)

At the moment we have submitted the eS, using social networks, Web sources and messengers, to a wide sample of 4555 citizens; among them 4512 (Table 1) agreed to participate. The participants could use their smartphone to access to the survey on the Web, on social networks and on messengers. The minimum age was 12 years; the maximum age was 85; the mean age was 49.3. The division by sex was: 2311 males 2201 females. 

### 3.2. Phone Submission (PS) Using a Wire Call 

At the moment, we have interviewed 1337 citizens; among them 1312 agreed (Table 1) to participate. The minimum age was 12 years; the maximum age was 84; the mean age was 48.9. The division by sex was: 642 males and 670 females. 

Both the *ES* and *PS* were conducted in Italy during the second wave of the COVID-19 pandemic from 30 November 2020 until 14 February 2021.

### 3.3. Methodological Flow

The methodological approach primarily involves submitting both *ES* and *PS* surveys.

After submission, a first important step will be based on the analysis of the two surveys on both samples to investigate any coherence or inconsistency through a robust statistical approach. This analysis will be based on some key elements, addressed in the survey, of the interaction with mobile technology in general (e.g., ability to use WhatsApp, App for social networks, generic Apps), of *mHealth* in general (medical Apps), of new *mHealth* tools (App for digital contact tracing) and digital health tools to interact with the health care processes.

The second step, after verifying the statistical coherence between the *ES* and the *PS*, will consist in focusing on the *ES* and investigating the same key elements on two subsamples of different ages (young and elderly subjects) to verify if the approach in the behavior towards technology depends on the age, however reporting a further validation of statistical significance with the corresponding subsamples of the *PS*.

We will follow two steps:(1)Verification of data normality.(2)Application of the T-student for the assessment of the coherence (not difference) with a *p* value higher than 0.1, when comparing *ES* and *PS*.

Application of the T-Student for the assessment of the significance of the difference with a *p* value lower than 0.01, when comparing the two groups different in age.

Regarding the statistical confidence interval of the investigated parameters, we set the goal of 95%.

Among the most used tests to verify if a distribution is approximate to a normal one are:

The Shapiro–Wilk test, which is preferable for small samples.

The Kolmogorov–Smirnov test, which instead is used for more numerous samples.

In consideration of the large amount of data, we opted for Kolmogorov–Smirnov.

## 4. Results and Discussion

### 4.1. General Outcome

The amount of data is large and further datamining will be required. Here, with aim of the article we present the outcome of the submission, comparing the two methods, *ES* and *PS*. The two methods, as shown below, report coherent and very similar results. We used the Student-*t* test to assess the statistics and fixed the lower limit to the acceptance of the H0 hypothesis (equality between averages) to *p* = 0.1.

To question 15 (Q15) (see Appendix A for the questions) “Do you have one or more smartphones?” 95.545% of the participants to the ES answered yes, while 94.981% of the participants to the PS answered yes (*no significance in the differences*; *p* = 0.198; Student-*t* test). Among the owners of smartphones, we then deepened the investigation relating to subsequent questions, classified as evaluation questions with a six-level psychometric scale; it was possible therefore to assign a minimum score of one and a maximum of six with, therefore, a theoretical mean value (TMV) of 3.5. The TMV can be referred to for comparison in the analysis of the answers. An average value of the answers below TMV indicates a more negative than positive response. An average value above TMV indicates a more positive than negative response.

Figure 2 shows the degree of familiarity with social network and messaging Apps (Q16–Q17) for both *ES* and *PS*. The results are very similar (*no significance in the differences*; *p* = 0.217, *p* = 0.237; Student-*t* test). For Apps which proved to be a lifebuoy during the pandemic, for maintaining special connections and as a support, including psychological support [5,6], the TMV >3.5 indicates a *more positive than negative* response showing a familiarity degree.

Figure 3 shows the familiarity with *Immuni*, the Italian App for the digital contact tracing (DCT) (Q18). Additionally, in this case the results were very similar for the two methods (*no significance in the differences*; *p* = 0.284; Student-*t* test) for a type of App which, in a pandemic period, is of vital importance for surveillance and monitoring. However, the TMV <3.5 indicates a *more negative than positive* response showing a low familiarity degree for a very strategic App.

For question Q19 on the familiarity of the other types of App there was a coherence in the results (*no significance in the differences*; *p* = 0.301; Student-*t* test), indicating a familiarity just above the threshold TMV (Figure 4).

Figure 5 reports the specific outcome with respect to aspects of digital health and with reference to *mHealth* (*no significance in the differences*; *p* = 0.333, *p* = 0.291; Student-*t* test) for both the ES and PS. The answers to the two questions indicate a value around the threshold for both the questions, with a value just below for the first question mainly related to the *mHealth* and just above for the other.

Surely what emerges from the analysis conducted through the graphs and the statistics referred to from time to time is important and can be highlighted with the following.

Firstly, from a general point of view, the two methodologies *ES* and *PS* had comparable performances as evidenced by the significance statistics. This was not taken for granted given that it is notorious that the methods of administration based on digital technologies present bias towards subjects affected by *Digital Divide*. 

The result certainly depends on the invitation present in the electronic survey for widespread dissemination and support in the compilation of those less accustomed to digital technology. 

This certainly leads us to highlight, in the case of *ES*, the presence of a sort of solidarity between subjects with a greater degree of ability in the digital towards those with a lower degree.

Specifically, if we consider the trend across the sample, we see how the following emerges in detail:In general, the apps for social networks and messaging (WhatsApp) are the most familiar. There is no doubt that in the pandemic era these have frequently represented a lifesaver for the population, to combat loneliness and any psychological consequences [5,6].A little familiarity and trust, all Italian, towards Immuni, an App for digital contact tracing, a problem that the stakeholder will then have to face.A low familiarity with *mHealth* Apps and this, in a pandemic time, where portable health could make a difference [4], is certainly another point that needs to be explored.A familiarity just above the threshold with regard to digital health processes in which we interact through digital health, which today have become essential for obtaining for example electronic prescriptions, for accessing blood tests and now also for vaccination.

On the one hand, the great role of mobile technology must be recognized without a doubt, as confirmed in the survey; on the other, it should be highlighted that, despite the strategic importance of the Apps in healthcare in this period, whether it is an App for digital contact tracing [7,8] for epidemiological monitoring, or an App for *mHealth* in general [4], the familiarity remains low.

This is certainly an important point on which to act strongly to reduce the Digital Divide in citizens of all the ages.

### 4.2. In-Depth Study in Two Sub-Samples: Young People and the Elderly against the Digital Divide

An important question to answer in this pandemic period is that of the generational relationship with *digital technology*. In other words, it is important to analyze how the DD acts between different generations. In this period, we have been bombarded by electronic surveys on various aspects related to the pandemic arriving through social networks, all surveys certainly affected by *bias*, as they did not adequately take into account the DD which also depends on familiarity with *digital technology* that the older ones do not have. With our two-way approach we tried to minimize this *bias*.

To obtain a first idea of DD based on age, we compared two samples in particular in two age groups in the *ES*: The first sample related to young people (aged 15–25 years): 413 subjects (201 males and 212 females), mean age 20.5 years, standard deviation 2.2; in what follows we refer to this sample as the young.The second sample related to elderly people (aged between 65 and 75 years): 382 subjects (199 females and 183 males), mean age 70.2 years, standard deviation 2.1; in what follows we refer to this sample with the term the elderly.

Many questions were asked using a graded psychometric scale (1 = minimum grade; 6 = maximum grade).

### 4.3. Intergroup Comparison

We report here the results of the *ES* for two very different age groups to analyze this question. We used the Student-*t* test to assess the statistics and fixed the higher limit to the acceptance of the H1 hypothesis (difference between averages) to *p* = 0.01. The comparison relative to the degree of familiarity of each parameter was made by comparing the corresponding parameter for young and elderly, and this was done parameter by parameter.

To question 15 (Q15) (see Appendix A for the questions) “Do you have one or more smartphones?”100% of the *young* subjects answered yes, while only 64.3% of the elderly answered yes (*high significance in the differences*, *p* = 0.009, Student-*t* test).

Among the owners of smartphones, we then deepened the investigation with the subsequent questions.

Figure 6 shows the degree of familiarity with social network and messaging apps (Q16–Q17) for young people and the elderly, from which there emerges a decidedly lower use by the elderly (*high significance in the differences*; *p* = 0.008, *p* = 0.007; Student-*t* test) of these Apps which proved to be a *lifebuoy* during the pandemic, for maintaining special connections and as a support, including as a psychological support [5,6].

Figure 7 shows familiarity with Immuni, an App for digital contact tracing (DCT) (Q18). Additionally, in this case the *young* people showed greater familiarity (*high significance in the differences*; *p* = 0.008, Student-*t* test) with a type of App which in a pandemic period is of vital importance for surveillance and monitoring. Additionally, to question Q19 on the familiarity of the other types of App there was a greater familiarity (*high significance in the differences*, *p* = 0.008, Student-*t* test) for *young* people than for the *elderly* (Figure 8).

Figure 9 reports the specific outcome with respect to aspects of digital health and to *mHealth*. From the answers relating to this battery of questions, it is evident that the DD of the elderly specifically related to the *Digital Health* appears very clearly (*high significance in the difference*; *p* = 0.008, *p* = 0.009; Student-*t* test).

The confidence interval of each parameter was always ≥95%.

The significance of the results was also validated through a further comparison of statistical significance between *ES* and *PS* with reference to these two subsamples. The answer to each question between *ES* and *PS* showed no significance in the differences (*no significance in the differences*, *p* lower limit of acceptance of H_0_, fixed to 0.1; Student-*t* test). 

Table 2 reports the two trends of the answers for both the groups. It shows a clear trend that, even for young people, there is a significant decrease in using or familiarity with Immuni, *mHealth*, and Digital Health. This trend is the same for elderly people. However, the elderly from what emerges in the table have a different tendency in dealing with these Apps and technologies. It is therefore important to analyze the differences in detail.

If we compare in detail the two groups, it can be observed that with regard to the use of mobile technology in general, while (a) in the use of the App for messaging and social media the average difference in the score is two (*p* = 0.007, Student *t* test), this difference (b) drops to 1.2 in the case of generic Apps (*p* = 0.008 Student *t* test), both due to a greater use of the latter by the elderly, with an average increase of 0.25 in the rating (equal to the 4.166%) and to a lower use of the latter by young people with an average decrease of 0.55 in the rating (equal to the 9.166%).

The difference (c) between the two groups in the use of the App for digital contact tracing is 1.9 as an average value and similar to the difference regarding the use of social and messaging Apps (*p* = 0.008 Student *t* test). However, in both groups there is an average decrease in the score, which in the elderly is equal to 0.95 (equal to he 15.83%) and in the young is equal to 1.05 (equal to 17.5%) clearly indicating a sharp decrease in familiarity with this essential App in the pandemic period.

If we look at the differences in behavior between the two groups in the use of apps for *mHealth* (d), we realize that between the two groups here the difference is narrower than the first comparison and is equal to 1.2 on average (*p* = 0.008; Studenti *t* test). This depends on a different way of interacting; in fact, young people with respect to the use of social and messaging apps show a very strong and a marked decrease in the degree of familiarity equal to 1.45 on average (equal to 24.1%); the elderly also show a decrease in familiarity, but this decrease is less and equal to 0.65 (equal to 10.83%). 

The difference between the two groups narrows further if we make the same comparison with the response relating to Digital Health.

Here, (e) the difference is only 0.7 as a mean value (*p* = 0.008; Student T test). The familiarity of young people with digital health processes drops by 1.65 on average (equal to 27.5%) if compared to that relating to social apps and WhatsApp. The elderly, on the other hand, making the same comparison lose less; in fact, the decrease recorded by them is 0.35 (equal to 5.83%).

What emerges from this analysis conducted through the comparison between the groups carried out with the aid of the previous graphs, Table 2 and the statistics applied from time to time is the following:An evident lower degree of familiarity on the part of the elderly with regard to mobile technology, *mHealth*, apps for DCT (Immuni) and digital processes.A great familiarity of young people with regard to mobile technology and in particular the social and messaging appsYoung people are less familiar with the App Immuni than with social and messaging apps.The gap in the score between young and old falls when considering *mHealth* and digital health. This presumably, is explained by a greater need on the part of the elderly to be connected to health processes whose digitization pushes them towards a forced familiarization. The last aspect shown in the table relating to digital health (in which the difference in score between young and elderly is only 0.7 of average) seems to highlight this: the elderly must connect with the health system for medical prescriptions, to obtain the results of the analyses, and now also to be vaccinated against SARS-CoV-2. However, this also highlights a loss of resources and opportunities, in fact young people, if they were adequately familiar with these technologies could for the elderly, also represent valid support here.

## 5. Conclusions 

### 5.1. Why the Need of a Study on the Digital Divide

The COVID-19 pandemic has created an unprecedented impetus for the development of *mHealth* [1,2,3,4]. This development involved both the enhancement and standardization of already consolidated solutions in *digital health* and the exploration of new potentials [2]. 

Numerous initiatives have been seen aimed at strengthening familiarity with digital technologies. 

Some initiatives have also been based on the development of surveys to thoroughly analyze the causes of the *Digital Divide* [13,14,15,16,17]. 

Other initiatives have focused on improvement initiatives on populations and/or ethnic groups that were disadvantaged from the start [18,19]. One thing is certain, the citizen during the pandemic and even subsequently will be increasingly called to interact with these technologies and above all with *mHealth*.

Self-care has proven to be a robust tool in times of the COVID-19 pandemic.

Important diseases such as diabetes [20] or cardiac diseases [21] benefit greatly from *mHealth* initiatives that rely on self-care and remote monitoring.

Even psychology and psychiatry, spurred on by new needs, have adapted during the pandemic to remote methodologies [22,23,24].

Further examples can be seen, without resulting in a review, in relation to new directions of digital health that are significantly increased at the moment, such as teleophthalmology [25] or totally new ones such as digital contact tracing [8,9].

There is no doubt that in the use of non-pharmaceutical technologies [26] will also significantly depend on overcoming cultural barriers.

It will be necessary to invest a lot of energy, also taking into account that the vaccination processes themselves, the most important resource of the moment, rely heavily on digital solutions, and that those who are already familiar with such technologies starting from multimedia tools [27] will find themselves able to cope better at this time. 

Following this reasoning, there is no doubt that, in many cases, the simple *mTech* itself has represented a *real lifebuoy* [5,6] both for the continuation of normal activities (working and teaching) and for providing a safety net. 

However, this has not always happened in a uniform way. The *digital divide* was a cause of this [9,10,11,12,13,14,15,16,17]. Making a map and an investigation of these aspects, is important both to consolidate experiences and to improve the diffusion of the medical technologies and their fruition. This depends on many factors ranging from difficulties in accessing instrumental resources to cultural and pathological problems (such as disabilities) [4]. Some studies have faced this using the surveys [13,14,15,16,17] focusing on particular factors and/or particular problems. An investigation of how these factors and problems act together is particularly important, and it is even more so in a pandemic period and was aimed in this study. 

### 5.2. What Has Been Proposed in the Study to Investigate the Digital Divide

A study on this issue conducted with purely multimedia technologies would pre-sent many limitations and above all it would risk excluding an important part of subjects affected by DD through bias.

To specifically address such an important problem, we prepared an articulated study and designed a survey that reached the largest number of subjects with an approach dedicated to the purpose. In particular we designed a survey that was submitted through two channels. 

The first channel was the electronic one and included a strong invitation to intra-family and intra-relationship sharing with the support of those with less familiarity with digital technologies. 

The second channel was based on interview contacts via the public wire telephone network.

### 5.3. What Are the Highlights from the Study?

The outcome of the study has five polarities.

A first polarity consists in having designed a methodology that allows the investigation of different aspects connected with the *Digital Divide* and that at the same time allows the estimation the possible impact of bias in wide-ranging surveys.

The second polarity consists of having verified, through an appropriate statistical approach, the consistency of the results of the two submission procedures (*ES* and *PS*). This indirectly showed us that the solutions made in the *ES* relating to an encouragement to share the survey with those less accustomed to digital technology have had a positive effect and that there was a sort of *intra-digital-cultural* solidarity.

A third polarity relative to the two *ES* and *PS* samples is a clear trend found in the sample. In particular, both in *ES* and *ES* with coherence it has been highlighted:The importance and the large use of and familiarity with the Apps for social networks and for messaging in coherence with other studies conducted in this period [5,6].A low familiarity with Digital Contact Tracing [7,8], that in Italy is based on the App Immuni [28].An unexpected low familiarity with *mHealth* Apps, and this worries us as this medical technology could make a difference [4]. This is certainly another point that needs to be explored.A familiarity just above the threshold with regard to digital health processes

The fourth polarity consists of a comparison between two subsamples of different ages to investigate the possible different degree of impact of the *Digital Divide* and therefore to verify if this characteristic depends on age. At the Istituto Superiore di Sanità, before the pandemic, we focused a lot on the problems of young people’s interaction with smartphones in general and therefore with apps and the Internet [29,30] and we have seen how this also leads to neuromuscular problems such as text neck [31,32] and psychological ones such as addiction [33]. However, we must note that this study seems to show us that in the face of this the young person with his ability to interact with these tools has a low degree of *Digital Divide* and an ability to interact with many Apps that during the pandemic have allowed and are allowing minimization of the sense of loneliness and the consequent related psychological problems.

Both an *intragroup* and an *intergroup* analysis were conducted on the two subsamples of different ages that highlighted: An evident lower degree of familiarity on the part of the elderly with regard to mobile technology, *mHealth*, Apps for DCT (Immuni) and digital processes.A great familiarity of young people with regard to mobile technology and in particular social and messaging apps, however not so great in the case of the App Immuni.The gap in the score between young and elderly falls when considering *mHealth* and digital health. This is particularly evident for Digital Health (in which the difference in score between the young and old is only 0.7 on average). This probably depends on the fact that the elderly, due to their having greater health problems, are more forced to interact with the medical digital processes for receiving analyses, online reservations, medical prescriptions and now also vaccines.

The fifth polarity that indirectly emerges from the study are the obvious suggestions for stake holders in relation to actions to:Minimize the digital divide in general from a general point of view and, in particular, where major criticalities have been highlighted, such as in *mHealth*.Minimize the digital divide on some categories, such as in this case the category of the elderly by acting in an incisive way on the range of wider problems.

Certainly, information, training and inter-generational sharing of knowledge will be fundamental.

### 5.4. What Are the Added Values of the Study?

From a general point of view the study presents four added values.

The first added value is the product (Appendix A) represented by the electronic survey tool that can be easily submitted through the *mTech* on the net during the pandemic. 

The second added value is represented by the survey which addresses in addition to the aspects of resources, such as those of the network [13], also other types of broad-spectrum problems.

The third added value is represented by the possibility of using this product, after minimal changes even in non-pandemic/post-pandemic periods.

The fourth added value is represented by the outcome with reference to: (a) the two methods *ES*, *PS* in the case of all the samples; (b) the two sub-samples in two age groups in the case of the *ES*, the first sample related to *young people* (aged 15–25 years), the second sample related to *elderly people* (aged 65–75 years). This outcome highlighted a DD strongly dependent on age and in particular a DD for the elderly particularly evident in the use of *mTech* in general and, in particular, in *mHealth* applications.

### 5.5. What the Study Supports and the Further Initiatives

From a general point of view, this article supports the initiatives that aim to reduce the *digital divide*, identifying the causes and bringing out potential solutions in a structured way, so that they can be submitted to stakeholders for a targeted and articulated approach.

Future developments of the study foresee, after adequate data-mining, an in-depth study of all the aspects proposed in the survey (Appendix A), from those relating to access to resources (e.g., the connection) up to those related to the training, disability and other cultural factors.

## Figures and Tables

**Figure 1 healthcare-09-00371-f001:**
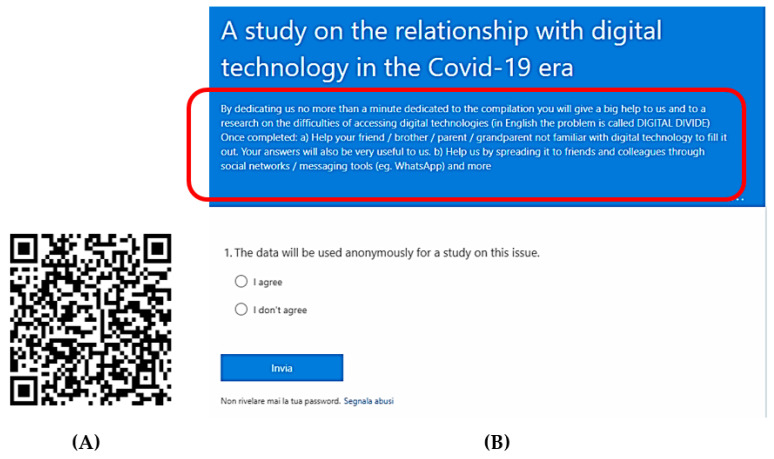
(**A**) The Quick Response code of the electronic survey (eS). (**B**) Request to support people who are not confident with the technology (e.g., Grandparents, parents).

**Figure 2 healthcare-09-00371-f002:**
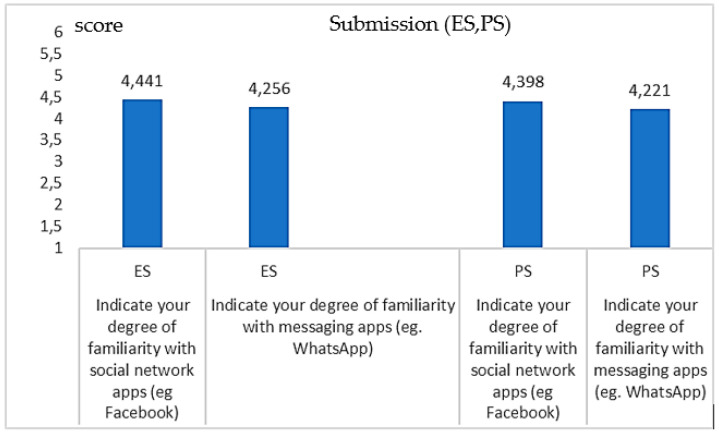
Coherence in the degree of familiarity with social networks, Q16 (*p* = 0.217) and messaging Apps, Q17 (*p* = 0.237) both for the ES and the PS. The graphic indicates a high coherence between the two groups.

**Figure 3 healthcare-09-00371-f003:**
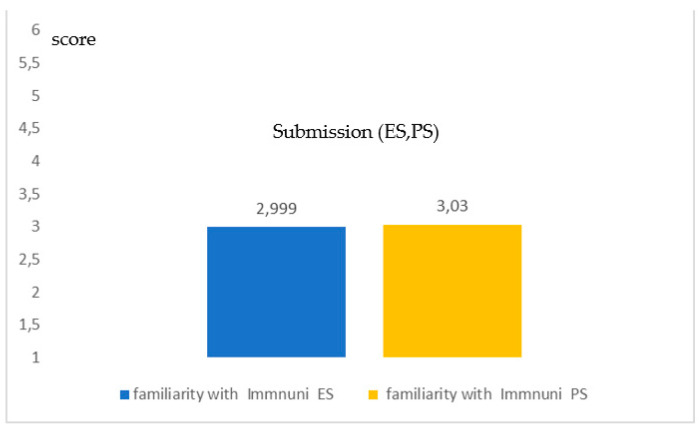
Coherence in the degree of familiarity with the App Immuni for digital contact tracing (DCT) both for the *ES* and the PS (*p* = 0.284). The graphic indicates a high coherence between the two groups.

**Figure 4 healthcare-09-00371-f004:**
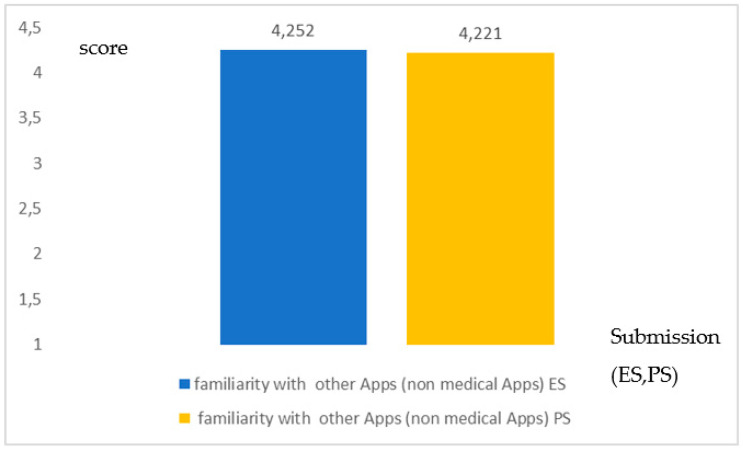
Coherence in the degree of familiarity with other Apps (non-medical Apps) both for the ES and the PS. (*p* = 0.301). The graphic indicates a high coherence between the two groups.

**Figure 5 healthcare-09-00371-f005:**
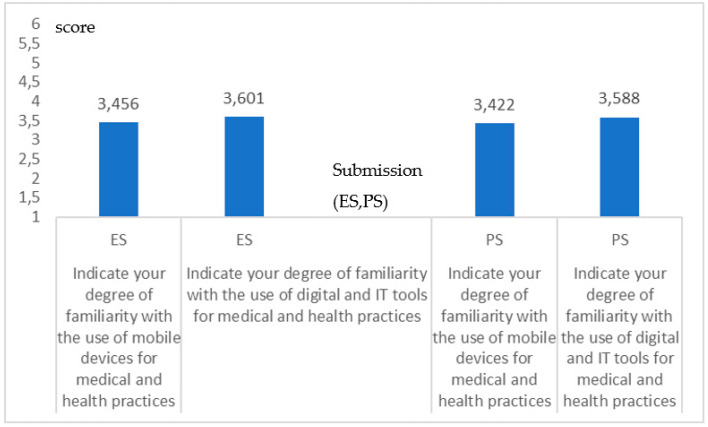
Coherence in the degree of familiarity assessment with *mHealth* (*p* = 0.333) processes and the Digital Health (*p* = 0.291) both for the *ES* and *PS. The graphic indicates a high coherence between the two groups.*

**Figure 6 healthcare-09-00371-f006:**
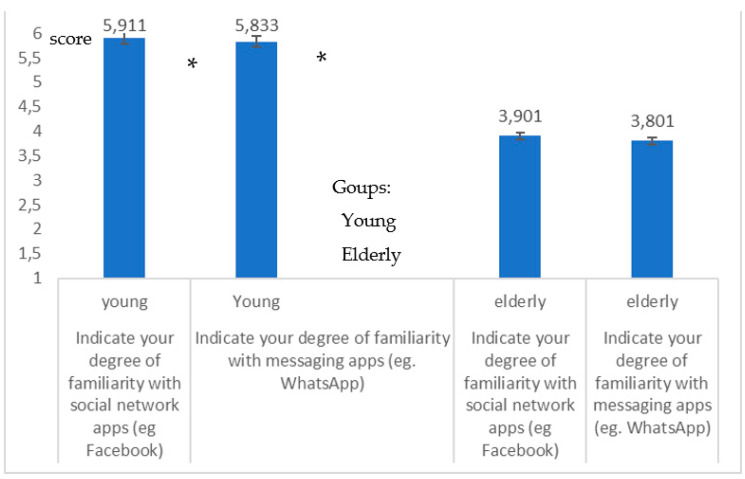
Difference in the degree of familiarity with social networks, Q16 and messaging apps, Q17 for young people and the elderly; the graphic indicates a high difference (*p* = 0.008, *p* = 0.007) between the two groups of young * and elderly.

**Figure 7 healthcare-09-00371-f007:**
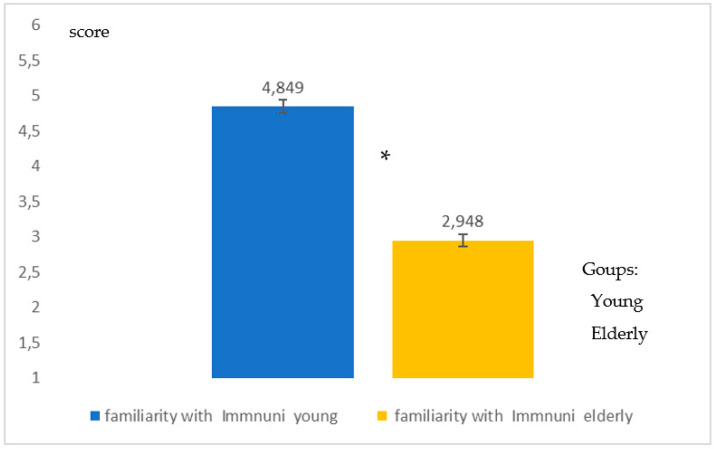
Difference in the degree of familiarity with the App Immuni for DCT for young people and the elderly. The graphic indicates a high difference (*p* = 0.008) between the two groups of young * and elderly people.

**Figure 8 healthcare-09-00371-f008:**
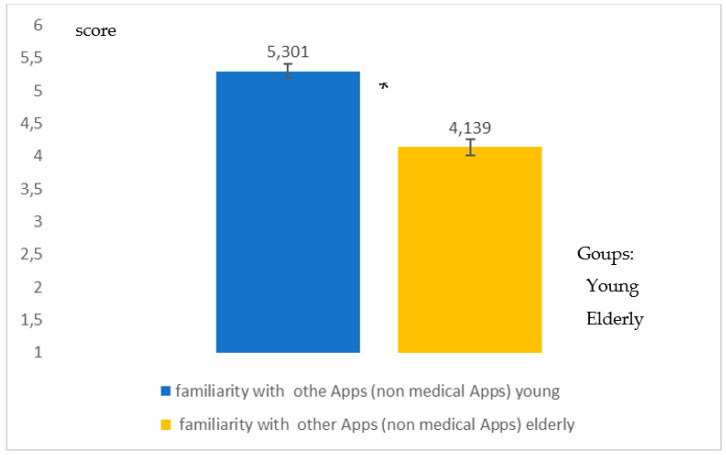
Differences in the degree of familiarity with other Apps (non-medical Apps) for young people and the elderly. The graphic indicates a high difference (*p* = 0.008) between the two groups of young * and elderly people.

**Figure 9 healthcare-09-00371-f009:**
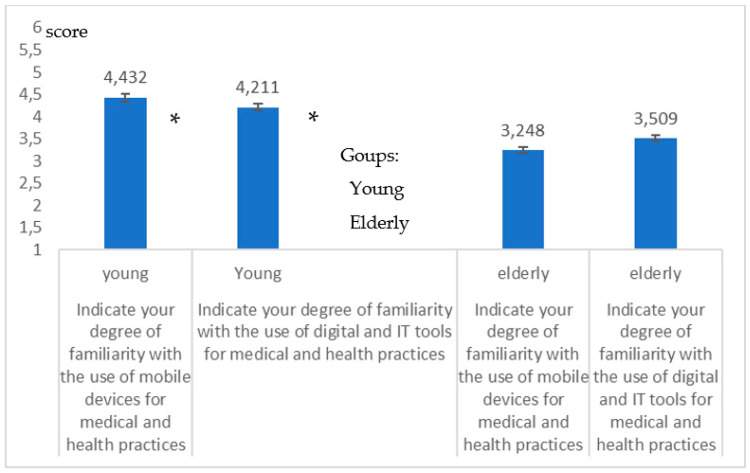
Differences in the degree of familiarity with *mHealth* (*p* = 0.008) processes and the *Digital Health* (*p* = 0.009) for young * people and the elderly.

**Table 1 healthcare-09-00371-t001:** Characteristics of the participants in the two submissions, electronic based on smartphone (*ES*) and telephonic (phone submission (*PS*)) based on a wire call.

Submission	Number Invited	Participants	Males/Females	Min Age/Max Age	Mean Age	Notes
*Electronic submission Using the smartphone*	4555	4512	2311/2201	12/85	49.3	No anomalies
*Phone (Wire call)*	1337	1312	642/670	12/84	48.9	No anomalies

**Table 2 healthcare-09-00371-t002:** Trends in the two groups.

	Social Media	Messaging	Immuni	Non-Medical Apps	*mHealth*	Digital Health
**Young**	5.9	5.8	4.8	5.3	4.4	4.2
**Elderly**	3.9	3.8	2,9	4.1	3.2	3.5

## Data Availability

The survey is available as a Appendix A.

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
