# Peer review of "The *Digital Divide* in the Era of COVID-19: An Investigation into an Important Obstacle to the Access to the *mHealth* by the Citizen"

_healthcare, 2021, doi:10.3390/healthcare9040371_

Round 1

Reviewer 1 Report

Thank you to the authors for the significant work on the paper. There are still some outstanding omissions and discrepancies in the paper.

Methods:

Please add a section to describe what analyses have been carried out (what can reader expect in results).

Results:

Phone submission participation was 1312 (page 5) people yet the results page 7 and 8 report very different figures if I am interpreting it correctly. Please address

Please add a table with a summary of the ages who responded to each type of survey and what technology devices and applications/websites they use.

Please present all numbers as number; percentage and confidence intervals (if appropriate)

Page 9: intragroup analyses: methodology should be added to methods. Please present number, percentage, confidence intervals and p values

All tables and figures need caption, x and y labels

Page 13: Intragroup comparison: I don’t understand what this means

It is difficult to comment on conclusion or discussion with the current discrepancies.

Author Response

Dear Reviewers and Editors,

Thank you very much for the reports you sent me which I found very useful and constructive.

From a general point of view on the basis of the suggestions and comments we have done the following actions:

  • Improvement of the presentation of data with tables and improvement of the clarity of the figures.
  • Improvement of the section dedicated to the comparison of the two groups starting from an analysis of the response trend, with a detailed numerical and statistical analysis.
  • Refine the discussion and conclusions by broadening it and refining the evidence from the study.
  • Revision of language and sentences and correction of typos.

ANSWERS TO REVIEWER 

Thank you to the authors for the significant work on the paper. There are still some outstanding omissions and discrepancies in the paper.

Methods:

Please add a section to describe what analyses have been carried out (what can reader expect in results).

The section “3.3 methodological flow” has been added with the text “The methodological approach primarily involves submitting both ES and PS surveys.

After submission, a first important step will be based on the analysis of the two surveys on both samples to investigate any coherence or inconsistency through a ro-bust statistical approach. This analysis will be based on some key elements, addressed in the survey, of the interaction with mobile technology in general (e.g. ability to use WhatsApp, App for social networks, generic Apps), of mHealth in general (medical Apps), of new mHealth tools (App for digital contact tracing) and digital health tools to interact with the health care processes.

The second step, after verifying the statistical coherence between the ES and the PS, will consist in focusing on the ES and investigating the same key elements on two subsamples of different ages (young and elderly subjects) to verify if the approach in the behavior towards technology depends on the age, however reporting a further validation of statistical significance with the corresponding subsamples of the PS

Results:

Phone submission participation was 1312 (page 5) people yet the results page 7 and 8 report very different figures if I am interpreting it correctly. Please address.

-There was a misalignment with the legends of the figures.

-It has now been corrected with the text “

to question Q19 on the familiarity of the other types of App there was a greater fa-miliarity (high significance in the differences, p=0.008, Student-t test) by young people than by the elderly (Figure 8).

Figure 9 reports the specific outcome with respect to aspects of digital health and  to mHealth.”

Please add a table with a summary of the ages who responded to each type of survey and what technology devices and applications/websites they use.

-Better clarifications have been added in the MS in several parts. See as not exhaustive example the modified title “3.2 Phone submission (PS) using a wire call” or the text in section 3.1 “The participants could use their smartphone to access to the survey on the WEB, on the social networks and on the messengers.”

-Table 1 has been added

Please present all numbers as number; percentage and confidence intervals (if appropriate)

-This has been applied in several points.

-See for example the text added in section 4.3 “If we compare in details the two groups it is observed that with regard to the use of mobile technology in general while (a) in the use of the App for messaging and social media the average difference in the score is 2 (p=0,007, Student T test), this difference (b) drops to 1.2 in the case of generic Apps (p=0,008 Student T test), both due to a greater use of the latter by the elderly, with an average increase of 0,25 in the rating (equal to the 4,166 %) and to a lower use of the latter by young people with an average decrease of 0.55 in the rating( equal to the  9,166 %).

 The difference (c) between the two groups in the use of the App for digital contact tracing is 1.9 in average value and similar to the difference with the use of the social and messaging Apps (p=0,008 Student T test). However, in both groups there is an average decrease in the score, which in the elderly is equal to 0.95 ( equal to the 15.83%) and in the young  is equal to 1.05 ( equal to 17.5%) clearly indicating a sharp decrease in familiarity with this App basic in the pandemic period.

 If we look at the differences in behavior between the two groups in the use of apps for mhealth (d) we realize that between the two groups here the difference is narrower than the first comparison and is equal to 1.2 in average value (p=0,008; Studenti T test). This depends on a different way of interacting, in fact young people with respect to the use of social and messaging apps show a very strong and marked decrease in the degree of familiarity equal to 1.45 in average value of the score (equal to 24.1%) ; the elderly also show a decrease in familiarity, but this decrease is less and equal to 0.65 (equal to 10.83%).”

Page 9: intragroup analyses: methodology should be added to methods. Please present number, percentage, confidence intervals and p values

-It has been done. See the answer to the first comment

-See the answer to a next comment

All tables and figures need caption, x and y labels

-It has been done

Page 13: Intragroup comparison: I don’t understand what this means

- The section has been joined with the section 4.3 and better clarified

-A wide analysis has been introduced and discussed.

-See the text added in this par.4.3 “Table 2, reports  the two trends of the answers for both the two groups. It shows a clear trend that, even for young people, there is a significant decrease in using or familiarity with Immuni, mHealth, and Digital Health. This trend is the same for elderly people. However, the elderly from what emerges in the table have a different tendency in dealing with these Apps and technologies. It is therefore important to analyze the differences in detail.

 If we compare ………………………………………………………………………………………………

 The difference (c) between the two groups in the use of the App for digital contact tracing is 1.9 in average value and similar to the difference with the use of the social and messaging Apps (p=0,008 Student T test). However, in both groups there is an average decrease in the score, which in the elderly is equal to 0.95 ( equal to the 15.83%) and in the young  is …………………………………………

Here (e) the difference is only 0.7 in mean value (p=0,008; Student T test). The familiarity of young people with digital

What emerges from this analysis conducted through the comparison between the groups carried out with the aid of the previous graphs, table 2 and the statistics ap-plied from time to time is the following:

  1. An evident …………………………………………………….

with the health system for medical prescrip-tion, to get the results of the analyzes, now also to get vaccinated from SARS-CoV-2. However, this also highlights a loss of resources and opportunities, in fact young peo-ple, if were ad-equately familiar with these technologies could for the elderly also here, represent a valid support”

It is difficult to comment on conclusion or discussion with the current discrepancies.

-After the suggested corrections we have refined these sections.

--See the text added in par.4.1 “Surely what emerges from the analysis conducted through the graphs and the sta-tistics referred to from time to time is important and can be highlighted with the fol-lowing………………………………………………………

Firstly, from a general point of view, the two methodologies ES and PS had com-parable performances as evidenced by the significance statistics.

  • A familiarity just above the threshold with regard to digital ealth pro-cesses in which we interact through digital health, which today have be-come essential for obtaining for example the electronic prescription, for accessing blood tests and now also for vaccination.

On the one hand, the great role of mobile technology must be recognized without a doubt, as confirmed in the survey; on the other, it should be highlighted that, despite the strategic importance of the Apps in healthcare in this period, whether it is an App for digital contact tracing [7,8] for epidemiological monitoring, or an App for mhealth in general [4] the familiarity remains low.

This is certainly an important point on which to act strongly to reduce the Digital Divide in the citizen of all the ages.”

-See the text added in par. 4.3 “Table 2, reports  the two trends of the answers for both the two groups. It shows a clear trend that, even for young people, there is a significant decrease in using or familiarity with Immuni, mHealth, and Digital Health. This trend is the same for elderly people. However, the elderly from what emerges in the table have a different tendency in dealing with these Apps and technologies. It is therefore important to analyze the differences in detail.

 If we compare ………………………………………………………………………………………………

 The difference (c) between the two groups in the use of the App for digital contact tracing is 1.9 in average value and similar to the difference with the use of the social and messaging Apps (p=0,008 Student T test). However, in both groups there is an average decrease in the score, which in the elderly is equal to 0.95 ( equal to the 15.83%) and in the young  is …………………………………………

Here (e) the difference is only 0.7 in mean value (p=0,008; Student T test). The familiarity of young people with digital

What emerges from this analysis conducted through the comparison between the groups carried out with the aid of the previous graphs, table 2 and the statistics ap-plied from time to time is the following:

  1. An evident …………………………………………………….

with the health system for medical prescrip-tion, to get the results of the analyzes, now also to get vaccinated from SARS-CoV-2. However, this also highlights a loss of resources and opportunities, in fact young peo-ple, if were adequately familiar with these technologies could for the elderly also here, represent a valid support.”

-See the text modified in par. 5.1 “It will be necessary to invest a lot of energy, also taking into account that the vaccina-tion processes themselves, the most important resource of the moment, are relying heavily on digital solutions ……………………. The digital divide was a cause of this [9-17]. Making a map and an investigation of these aspects, is important both to consolidate experiences and to improve the diffusion of the medical technolo-gies and their fruition.  This depends on many factors ranging from difficulties in ac-cessing instrumental resources to cultural and pathological problems (such as disabili-ties) [4]. Some studies have faced this using the surveys [13-17] focusing on particular factors and /or particular problems. An investigation of how these factors and prob-lems act together is particularly important, and it is even more so in a pandemic period and was aimed in this study.”

See the text added in par. 5.2 “A study on this issue conducted with purely multimedia technologies would pre-sent many limitations and above all it would risk excluding an important part of subjects affected by DD through bias.

To specifically address such an important problem, we have prepared an articu-lated study and designed a survey that reached the largest number of subjects with an approach dedicated to the purpose.  In particular we designed a survey that was submitted through two channels.

The first channel was the electronic one, and included a strong invitation to in-tra-family and intra-relationship sharing with the support of those with less familiari-ty with digital technologies.

The second channel was based on interview contacts via the public wire telephone network.”

See the text added in par. 5.3 “The outcome of the study has five polarities.

A first polarity consists in having designed a methodology that allows to investi-gate different aspects connected with the Digital Divide and that at the same time al-lows to estimate the possible impact of bias in wide-ranging surveys………………”

Reviewer 2 Report

I accept changes made by Authors and their anwsers to my review.

Author Response

Dear Reviewers and Editors,

Thank you very much for the reports you sent me which I found very useful and constructive.

From a general point of view on the basis of the suggestions and comments we have done the following actions:

  • Improvement of the presentation of data with tables and improvement of the clarity of the figures.
  • Improvement of the section dedicated to the comparison of the two groups starting from an analysis of the response trend, with a detailed numerical and statistical analysis.
  • Refine the discussion and conclusions by broadening it and refining the evidence from the study.
  • Revision of language and sentences and correction of typos.

I accept changes made by Authors and their anwsers to my review.

-Thank you for the encouraging comment

Reviewer 3 Report

Executive Summary:

The resubmitted manuscript titled “The Digital Divide in the Era of Covid-19: an Investigation into an Important Obstacle to the Access to the mHealth by the Citizen” investigated the phenomenon of the digital divide in Italy during the Covid-19 pandemic. It focuses on the availability of smartphones and digital healthcare services. In general, the quality of this manuscript improved compared to the previous rejected version. However, authors still need to perform major revisions to meet the requirement of publishing.

Major Comments:

  1. Discussion and Conclusion:

In section “4. Results and Discussion”, the authors listed all the results but did not perform a detailed discussion regarding DD. In section “5. Conclusion”, the authors provided too many non-conclusion writings. I recommend authors to re-write the discussion and conclusion. This will significantly improve the quality of the manuscript.

  1. Section “4.4. Intra-group comparison”

Section 4.4 is in response to my peer review. However, the authors basically copy and paste my comments, including my table. The goal for me to provide this table is to provide a hint for authors to think about. The goal is to find out whether people’s access to digital health apps is “age-related”. Therefore, the authors need to perform statistical analysis to find out the “real” result of the inter-group trend with proper discussion.

Minor Comments:

  1. In all column figures, please use asterisks for significantly different groups.
  2. P-value and group information should be written under designated figures. Please refer to other publications in Healthcare for your information.

Author Response

Dear Reviewers and Editors,

Thank you very much for the reports you sent me which I found very useful and constructive.

From a general point of view on the basis of the suggestions and comments we have done the following actions:

  • Improvement of the presentation of data with tables and improvement of the clarity of the figures.
  • Improvement of the section dedicated to the comparison of the two groups starting from an analysis of the response trend, with a detailed numerical and statistical analysis.
  • Refine the discussion and conclusions by broadening it and refining the evidence from the study.
  • Revision of language and sentences and correction of typos.

ANSWERS TO REVIEWER 

 Executive Summary:

The resubmitted manuscript titled “The Digital Divide in the Era of Covid-19: an Investigation into an Important Obstacle to the Access to the mHealth by the Citizen” investigated the phenomenon of the digital divide in Italy during the Covid-19 pandemic. It focuses on the availability of smartphones and digital healthcare services. In general, the quality of this manuscript improved compared to the previous rejected version. However, authors still need to perform major revisions to meet the requirement of publishing.

Major Comments:

  1. Discussion and Conclusion:

In section “4. Results and Discussion”, the authors listed all the results but did not perform a detailed discussion regarding DD. In section “5. Conclusion”, the authors provided too many non-conclusion writings. I recommend authors to re-write the discussion and conclusion. This will significantly improve the quality of the manuscript.

-The discussion and conclusions have been completely revised.

-See the architectural changes made in par.4 and par. 5

-See the text added in par.4.1 “Surely what emerges from the analysis conducted through the graphs and the sta-tistics referred to from time to time is important and can be highlighted with the fol-lowing………………………………………………………

Firstly, from a general point of view, the two methodologies ES and PS had com-parable performances as evidenced by the significance statistics.

  • A familiarity just above the threshold with regard to digital ealth pro-cesses in which we interact through digital health, which today have be-come essential for obtaining for example the electronic prescription, for accessing blood tests and now also for vaccination.

On the one hand, the great role of mobile technology must be recognized without a doubt, as confirmed in the survey; on the other, it should be highlighted that, despite the strategic importance of the Apps in healthcare in this period, whether it is an App for digital contact tracing [7,8] for epidemiological monitoring, or an App for mhealth in general [4] the familiarity remains low.

This is certainly an important point on which to act strongly to reduce the Digital Divide in the citizen of all the ages.”

See the text added in par. 4.3 “Table 2, reports  the two trends of the answers for both the two groups. It shows a clear trend that, even for young people, there is a significant decrease in using or familiarity with Immuni, mHealth, and Digital Health. This trend is the same for elderly people. However, the elderly from what emerges in the table have a different tendency in dealing with these Apps and technologies. It is therefore important to analyze the differences in detail.

 If we compare ………………………………………………………………………………………………

 The difference (c) between the two groups in the use of the App for digital contact tracing is 1.9 in average value and similar to the difference with the use of the social and messaging Apps (p=0,008 Student T test). However, in both groups there is an average decrease in the score, which in the elderly is equal to 0.95 ( equal to the 15.83%) and in the young  is …………………………………………

Here (e) the difference is only 0.7 in mean value (p=0,008; Student T test). The familiarity of young people with digital

What emerges from this analysis conducted through the comparison between the groups carried out with the aid of the previous graphs, table 2 and the statistics ap-plied from time to time is the following:

  1. An evident …………………………………………………….

with the health system for medical prescrip-tion, to get the results of the analyzes, now also to get vaccinated from SARS-CoV-2. However, this also highlights a loss of resources and opportunities, in fact young peo-ple, if were adequately familiar with these technologies could for the elderly also here, represent a valid support.”

-See the text modified in par. 5.1 “It will be necessary to invest a lot of energy, also taking into account that the vaccina-tion processes themselves, the most important resource of the moment, are relying heavily on digital solutions ……………………. The digital divide was a cause of this [9-17]. Making a map and an investigation of these aspects, is important both to consolidate experiences and to improve the diffusion of the medical technolo-gies and their fruition.  This depends on many factors ranging from difficulties in ac-cessing instrumental resources to cultural and pathological problems (such as disabili-ties) [4]. Some studies have faced this using the surveys [13-17] focusing on particular factors and /or particular problems. An investigation of how these factors and prob-lems act together is particularly important, and it is even more so in a pandemic period and was aimed in this study.”

-See the text added in par. 5.2 “A study on this issue conducted with purely multimedia technologies would pre-sent many limitations and above all it would risk excluding an important part of subjects affected by DD through bias.

To specifically address such an important problem, we have prepared an articu-lated study and designed a survey that reached the largest number of subjects with an approach dedicated to the purpose.  In particular we designed a survey that was submitted through two channels.

The first channel was the electronic one, and included a strong invitation to in-tra-family and intra-relationship sharing with the support of those with less familiari-ty with digital technologies.

The second channel was based on interview contacts via the public wire telephone network.”

-See the text added in par. 5.3 “The outcome of the study has five polarities.

A first polarity consists in having designed a methodology that allows to investi-gate different aspects connected with the Digital Divide and that at the same time al-lows to estimate the possible impact of bias in wide-ranging surveys………………”

  1. Section “4.4. Intra-group comparison”

Section 4.4 is in response to my peer review. However, the authors basically copy and paste my comments, including my table. The goal for me to provide this table is to provide a hint for authors to think about. The goal is to find out whether people’s access to digital health apps is “age-related”. Therefore, the authors need to perform statistical analysis to find out the “real” result of the inter-group trend with proper discussion.

-The section has been joined with the section 4.3.

-A wide analysis has been introduced and discussed.

-See the text added in this par. Reported in the answer to the previous comment                        “

Minor Comments:

  1. In all column figures, please use asterisks for significantly different groups.
  2. P-value and group information should be written under designated figures. Please refer to other publications in Healthcare for your information.

-We have proposed and revised the presentation of these figures and are open to editorial changes during the process using powerpoint or other tools allowed by the journal

Round 2

Reviewer 1 Report

Thanks to the authors for their improvements.

Some specific suggestions:

Methods: In the methodological flow, please include details of statistical tests that follow in the results rather than detail in results. What coherence score was used? What was the confidence interval for the score?

What tests were used for the intergroup comparison? Please include reasoning for test selection in methods.

Unsure what ‘difference in degree of familiarity variable’: Is it the mean degree of familiarity for each age group? please explain how calculated in methods.

Figure 6: Is the first ‘young’ refer to PS and the second ‘young to ‘ES’. Please annotate diagram to clarify. Same with figure 7 and 8. Please include error bars on diagrams and confidence intervals in text.

The discussion would benefit from integration with existing literature to put the study findings in context. E.g. published studies the relationship between uptake of health apps and age. Is there stigma about age related use of the internet?

Author Response

Dear Reviewers and Editors,

Thank you very much for the reports you sent me during the peer review.

Reviewer 1 Thanks to the authors for their improvements. Some specific suggestions:

Thank you for these further suggestions

 Methods: In the methodological flow, please include details of statistical tests that follow in the results rather than detail in results. What coherence score was used? What was the confidence interval for the score?

We have now explained this. See the added text  in the methods“Regarding the statistics in the investigated parameters. We will follow two steps:1) verification of data normality 2) Application of the T-student for the assessment of the coherence (not difference) with a pvalue higher that 0.1, when comparing ES and PS. Application of the T-Student for …………….. of the difference with a pvalue lower than 0.01, when comparing the two groups different in age……..In consideration of the large number of data we opted for Kolmogorov-Smirnov

What tests were used for the intergroup comparison? Please include reasoning for test selection in methods.

We have explained this. Se the answer to the previous comment

Unsure what ‘difference in degree of familiarity variable’: Is it the mean degree of familiarity for each age group? please explain how calculated in methods.

We have inserted a sentence to clarify this. See the added text “The comparison relative to the degree of familiarity of each parameter was made by comparing the corresponding parameter for young and elderly, and this was done parameter by parameter.”

Figure 6: Is the first ‘young’ refer to PS and the second ‘young to ‘ES’. Please annotate diagram to clarify. Same with figure 7 and 8. Please include error bars on diagrams and confidence intervals in text.

No, they are both referred to ES. We have added the information on the confidence and the error bars in these diagrams.

The discussion would benefit from integration with existing literature to put the study findings in context. E.g. published studies the relationship between uptake of health apps and age. Is there stigma about age related use of the internet?

We have added the text “At the Istituto Superiore di Sanità, before the pandemic, we focused a lot on the prob-lems of young people's interaction with smartphones in general and therefore with apps and the internet [29-30] and we have seen how this also leads to neuromuscular problems such as the text neck [31-32] and psychological ones such as addiction [33]. However, we must take note that this study seems to show us that in the face of this the young person with his ability to interact with these tools, has a low degree of digital divide and an ability to interact with many Apps that during the pandemic have allowed and are allowing to minimize the sense of loneliness and the consequent related psychological problems.

And added  Ref. 29-33

Reviewer 3 Report

The current version of the manuscript provides high quality of research, discussion, and conclusion. I recommend accepting in the current form. Congratulations to all authors for their effort.

Author Response

Dear Reviewers and Editors,

Thank you very much for the reports you sent me during the peer review.

Reviewer 3.

Thank you for your encouraging comments

This manuscript is a resubmission of an earlier submission. The following is a list of the peer review reports and author responses from that submission.

Round 1

Reviewer 1 Report

Overall comment

In this study the authors are investigating the challenges with using technology during COVID-19. I appreciate the intent of the authors, however, I have some serious concerns about the study design data collection and analysis.

My main concerns are:

Designing an online survey to understand accessibility issues to online media seems unlikely to reach population of people who don’t routinely use online media, which as I understand it are the people you need to reach to answer your research question

Sampling study data by age for analysis and then reporting age related differences as the main finding is problematic for me. It lacks objectivity.

The writing often has long sentences. The phrasing is sometimes difficult to understand. As an example: “ With the aim of proposing a survey on this area, to make a map point and investigate the problem”: unsure what map point is; sentence is repetitive and should be rephrased

Results should be separated from methods. The methods section require detail on design. I would suggest the authors look at guidelines for reporting observational cross sectional surveys e.g. STROBE and incorporate this structure and rigour to reporting. Possible to include the survey in an appendix? The results should be presented from the entire survey rather than presenting subgroup analysis.

Reviewer 2 Report

The paper contains numerous shortcomings. Firstly, the goal of the paper has not been clearly defined.  The paper contains “Purpose” section, but it is devoted to the survey preparation. Also literature analysis is very limited based on 13 references only. There is no information about sampling methodology used in the research. Because of this it is not clear if the sample is representative or not. The paper contains no clear description of research problem, research questions and research hypotheses. Presentation of the research results is superficial (section “Protocol and Results”). The same remark relates to the “Discussion and Conclusion” section.

Reviewer 3 Report

Executive Summary

The manuscript titled “The Digital Divide in the Era of Covid-19: an Investigation into an Important Obstacle to the Access to the mHealth by the Citizen” described a digital divide issue in Italy, which also applies to many other countries in the world. The outbreak of the Covid-19 pandemic requires citizens to use tracking apps and medical apps to access healthcare services in a contact-free manner. However, due to the digital divide, these apps may not apply to some subgroups of citizens. Overall, this topic is very practical and important. However, due to the design and data analysis, the authors could not fully support the conclusion. Therefore, my suggestion for this manuscript is a major revision.

Major Comments

  • Survey method

Under section “3. Protocol and Results”, the authors explained the difficulties of using paper-based survey tools. I agree with that. Unfortunately, the topic of this research is the influence of the “digital divide”, which means the divide between digital tools and paper tools. With this survey setting, an unconscious bias formed from the results. People who do not have access to digital tools, which is important in this research, are automatically excluded.

  • Results and data analysis

Please indicate the statistical methods you used in this manuscript, especially the ones to calculate P-value.

  • Figure 2 to Figure 5

From Figure 2 to Figure 5, the authors compared the answers between “Young” and “Elderly”. This is an intergroup comparison. However, to have a scientific conclusion, an intragroup comparison is also needed. For example:

Social Media

Messaging

Immuni

Non-Medical Apps

mHealth

Digital Health

Young

5.9

5.8

4.8

5.3

4.4

4.2

Elderly

3.9

3.8

2.9

4.1

3.2

3.5

Please pay attention to each group. You can see a clear trend that, even for young people, there is a significant decrease in using Immuni, mHealth, and Digital Health. This trend is the same for elderly people. However, the elderly people have a higher ratio of using Immuni, mHealth, and Digital Health. In another word, young many young people who have the technology did not use Immuni, mHealth, and Digital Health. This is very important since there is a waste of digital resources from young people while elderly people are behaving differently. These results need to be discussed to show a comprehensive conclusion.

Minor Comments

The questions in the survey need to be better designed. For example, in question 6, the answers are not excluding each other. For example, a student can have a part-time job. And, a person who is currently unemployed due to a pandemic can be actively searching for a job. Further, “I am not retired” does not provide much value in this survey.